# Effects of Governmental Intervention on Foodborne Disease Events: Evidence from China [note 1]

**DOI:** 10.3390/ijerph182413311

**Published:** 2021-12-17

**Authors:** Zhuang Zhang, You-hua Chen, Lin-hai Wu

**Affiliations:** 1College of Economics and Management, South China Agricultural University, Guangzhou 510642, China; 20192001006@stu.scau.edu.cn; 2Research Center for Green Development of Agriculture, South China Agricultural University, Guangzhou 510642, China; 3School of Business, Jiangnan University, Wuxi 214122, China; wyy@jiangnan.edu.cn

**Keywords:** foodborne disease events, governmental intervention, temporal and spatial distribution, health equity, I12, Q18

## Abstract

Foodborne disease events (FDEs) endanger residents’ health around the world, including China. Most countries have formulated food safety regulation policies, but the effects of governmental intervention (GI) on FDEs are still unclear. So, this paper purposes to explore the effects of GI on FDEs by using Chinese provincial panel data from 2011 to 2019. The results show that: (i) GI has a significant negative impact on FDEs. Ceteris paribus, FDEs decreased by 1.3% when government expenditure on FDEs increased by 1%. (ii) By strengthening food safety standards and guiding enterprises to offer safer food, government can further improve FDEs. (iii) However, GI has a strong negative externality. Although GI alleviates FDEs in local areas, it aggravates FDEs in other areas. (iv) Compared with the eastern and coastal areas, the effects of GI on FDEs in the central, western, and inland areas are more significant. GI is conducive to ensuring Chinese health and equity. Policymakers should pay attention to two tasks in food safety regulation. Firstly, they should continue to strengthen GI in food safety issues, enhance food safety certification, and strive to ensure food safety. Secondly, they should reinforce the co-governance of regional food safety issues and reduce the negative externality of GI.

## 1. Introduction

Currently, foodborne diseases (FDEs) are a vital and universal issue [1,2]. Almost 1 in 10 people worldwide fall sick each year from eating contaminated food [3]. FDEs lead to 42,000 deaths every year, especially in Africa and Southeast Asia [3]. The situation in China is also serious, with an average of 1 in 6.5 people suffering from diseases due to the ingestion of food contaminated by foodborne pathogens [4]. Residents’ food safety and health have been challenged during the COVID-19 epidemic. Preventing FDEs is an important task for all countries to ensure national security.

Governments, media, and consumers are vital participants in controlling FDEs [5]. The media can provide production information on food companies, promote consumers’ understanding of the products, and alleviate the information asymmetry between consumers and producers [5,6]. The education, knowledge, and economic status of consumers affect the occurrence of FDEs [7,8,9,10]. Among all participants, the government has the most decisive authority in food supervision. For example, the government can affect food safety in production, circulation, and sales by means of enacting food safety laws, issuing administrative orders, and implementing economic penalties (which can be called government intervention (GI)) to affect the outbreak of FDEs [10,11,12]. However, the effects of GI are always controversial. Some scholars emphasize the positive impact of GI on innovation and enterprise performance. For example, using a case analysis, Porter et al. demonstrated that suitable GI can drive enterprises to improve production performance [13]. Wang et al. found beneficial effects of GI on innovation in Hong Kong and Singapore [14]. Fang et al. declared that GI played a significant role in fighting COVID-19 [15]. Some traditional conceptions hold that GI will fail because the regulated firm can bribe regulators if the governance system is not perfect [16,17,18]. Recent studies also provide new evidence that GI aggravates a firm’s investment misallocation and deteriorate innovation efficiency [19,20]. There may be similar conclusions about the effect of GI on FDEs.

In China, the current situation of FDEs and food safety is not optimistic. According to the global food security index (GFSI), China’s food security ranked 35th among 113 countries in 2019 [21]. Therefore, Chinese policymakers need to pay more attention to food safety. Since the tainted milk powder incident in 2008, the Chinese government has devoted considerable efforts to food regulation [5,10,11]. In 2009, China promulgated the Food Safety Law to strictly rectify the bad production behaviors of enterprises [10]. In 2013, China streamlined administration and delegated powers to food supervision, highlighting the need for adequate control of food production. In 2015, China promulgated the second edition of the Food Safety Law to ensure food safety. However, there is still debate about the impact of GI on food safety. Some studies believe the improvement of the social food safety legal system is conducive to further ensuring food safety [22], especially in China, where consumers prefer food certified by the government [23]. Therefore, strengthening governmental intervention is vital to ensuring food safety and improving corporate performance. However, some evidence shows that Chinese GI may not affect food safety. According to Chinese official media [24], China’s food safety standards have been in line with those of the rest of world and are even higher than global standards. Although milk powder is now in line with global standards, its demand by Chinese parents is still low. This shows that GI may not affect food supply and market demand. Chu also found that Chinese food safety laws significantly affect imported food more than domestic food [25].

Although numerous studies have explored the relationship between government intervention (GI) and foodborne disease events (FDEs) or food safety via theoretical analysis [10], case analysis [25], and linear regression [26], the relationship is still ambiguous. However, knowing the mechanism and efficiency of GI on FDEs has significant theoretical and practical significance. So, this article further considers the mechanism of GI on FDEs. Moreover, considering the spatial characteristics of FDEs, this study tries to capture the effect of GI on FDEs via a spatial econometric model (because the temporal and spatial correlation of FDES in China is very obvious, see Figure 1). Our research shows that the impact of GI on FDEs critically depends on its negative externality. Only when the negative externality is strong enough will GI have a significant negative impact on FDEs. This study provides a valuable guidance for the Chinese government about changing the supervision approach, strengthening regional food supervision cooperation, ensuring residents’ food safety, and controlling FDEs.

## 2. Methodology

### 2.1. Data

To analyze the spatiotemporal characteristics and vital socioeconomic factors of FDEs, panel data were collected from the China Health Statistics Yearbook, China Statistical Yearbook, National Meteorological Science Data, and China Green Food Development Center. The data range from 2011 to 2019, covering 30 provincial administrative regions of China (27 provinces and 4 municipalities directly under the central government in China mainland, excluding Tibet). All the data were processed manually.

The China Health Statistics Yearbook offered us details about the number of foodborne disease events (FDEs) and health supervision (center) personnel. The China Statistical Yearbook provided us with details about per capita GDP, consumer price index, urban and rural disposable income, urban and rural population, and corresponding information on education (illiteracy, primary school, junior high school, senior high school, junior high school, university, and graduate students). The National Meteorological Science Data (dataset of daily surface climatological data over China (V3.0)) offered us details about sunshine, temperature, and rainfall in each province. The China Green Food Development Center offered us details about green food certification and labeling information in each province.

### 2.2. Variables

(1)Dependent variable: FDEs. The statistical number of FDEs in the China Health Statistics Yearbook comes from the National Health and Family Planning Commission, the State Administration of Traditional Chinese Medicine (SATCM), and SATCM’s subordinate medical system. To the best of our knowledge, these data are the most authoritative.(2)Independent variable: GI. According to the existing research, it is not easy to obtain the indicators related to the investment of direct supervision of FDEs. Compared with other indicators, the number of people in health supervision (center) relevantly reflects the government’s efforts for controlling FDEs. Similarly to Zhang et al. [12], we calculated the likely public fiscal expenditure on FDEs as an indicator of government intervention.
(1)$$GI=\frac{Number\;of\;people\;in\;health\;\sup ervision}{Number\;of\;total\;medical\;staff}\times medical\;expenditure$$(3)Control variables: existing studies show that the natural environment and socioeconomic factors have a crucial impact on the occurrence of FDEs [27,28,29,30]. In general, FDEs in developed areas are affected by socioeconomic factors more seriously, while in less-developed areas, FDEs are heavily affected by the natural environment. Michael et al. stressed the influence of economic and social factors on food safety and risk [31]. People believe that food safety in an area is mainly affected by five factors: education, social network, social capital, family income, and unemployment rate [31]. Explicitly speaking, the different socioeconomic factors, per capita GDP, urbanization, inequality, price, and education play vital roles in FDEs. Of course, uncertainty is also a reasonable cause of FDEs [32,33]. According to the availability of data, the following control variables were selected:
(a)Economic growth. Studies show a positive correlation between per capita GDP and FDEs [5,34]. However, an inverted U-shape relationship has also been found [26]. In the early stages of economic development, to achieve food security, many countries neglected the food production supervision process [26,35]. However, as the per capita income of a country increases, consumers’ demand for low-end food decreases significantly, while their need for food quality rises significantly [36,37,38]. Then, the whole society is required to ensure food quality with a good food regulatory system. Economic growth was measured by per capita GDP because per capita GDP reflects economic growth more accurately than the total GDP [12].(b)Education. Education’s effect on FDEs is mainly reflected in three aspects. First, people with higher education have a richer knowledge about food safety and pay more attention to it [7]. Second, people who have received higher education can share their knowledge of food storage, food nutrition, and health with other residents. They can prevent FDEs by teaching others to check the food instructions or texture. For example, parents’ food safety knowledge affects their children’s knowledge [8]. Third, education helps spread food safety knowledge to a wider range of people via new media. The Internet and social media can amplify the effect of food safety education and spread this information to many people [39].Education is an essential factor affecting the acquisition and dissemination of food safety knowledge. Based on the research of Zhang et al. [12], we calculated the average years of education according to the proportion of the population with different education levels. That is, average years of education = (0 × Illiteracy + 6 × Number of primary school students + 9 × Junior high school + 12 × Number of senior high school students + 15 × Number of junior college students + 16 × Number of university students + 19 × Number of graduate students)/total number of populations.(c)CPI. Price is related to product quality. High prices may mean higher quality [35] and may decrease the occurrence of FDEs. However, a general rise in prices will lead to an increase in food costs for some residents. This leads, in turn, to consumers’ searching for low-quality food without an explicit safety guarantee, which increases the occurrence of FDEs. However, the Chinese price control system is relatively strict, and the impact of price on FDEs is not significant. To reflect the consumer’s response in the market, CPI (consumer price index) is used to measure market price changes.(d)Urbanization. The impact of urbanization on FDEs is uncertain. Compared with rural areas, the urban food supervision system is more effective, and urban areas experience less occurrences of FDEs [40]. However, urban people prefer to eat in restaurants or roadside stalls, and food safety within these establishments is often not guaranteed [41]. The proportion of registered residence population and total population in the city was used to measure urbanization.(e)Environmental indicators. Environment is also a key variable for FDEs. Adane et al. found that the environment, particularly regarding high humidity and high temperature, is conducive to fungal reproduction, resulting in severe FDEs in sub-Saharan areas [27]. Recent studies show that temperature may affect FDEs by influencing the choice of dining place: eating at home or eating out [42]. However, a study on England and Wales found that the impact of temperature on FDEs decreased as time went by [43]. Based on the Barnes method [44], the IDW (inverse distance weighted) method is used to interpolate the grid data. It covers China’s 500 × 500 grid, with each grid size at 0.1231924 (longitude) × 0.994549 (latitude). Then, China’s annual average sunshine, rainfall, and temperature data from 2011 to 2019 can be calculated.(f)Theil index. Inequality indicates the allocation of natural or social resources in society. For example, in a country where the poor account for the vast majority of the population, most people do not have enough money to buy safe food or enough resources to produce food [45]. In such counties, there is a high frequency of FDEs. Inequality affects the availability and affordability of food in a country or region [46]. Availability and affordability are the vital ways that income inequality affects FDEs. The Theil index can reflect the urban–rural population gap and income gap and capture the impact of urban–rural inequality on FDEs. The Theil index has a reasonably consistent Gini index, and it is easier to decompose and reflect the inequality of different dimensions [47]. Moreover, compared with the Gini coefficient, the Theil index not only reflects economic differences but also comprehensively considers demographic factors. It is calculated as:
(2)$$Theil=\sum_{i=1}^{2}\left(\frac{I_{i}}{I}\times \ln(\frac{I_{i}/I}{P_{i}/P})\right)$$In Equation (2), I denotes the total income of urban and rural areas, $I_{i}$ denotes the sum of the urban and rural population, $I_{i}$ represents the income in urban or rural areas, $P_{i}$ denotes the population of urban or rural areas. i=1 denotes urban areas, while i=2 denotes rural areas.(4)Mechanism variable: food safety. The research showed that “green food” and organic food can ensure the health of residents and reduce foodborne diseases [48]. So, if a firm has more “green food certifications” or “green food labels”, it produces safer and higher-quality foods [48]. Therefore, we used the average number of green food certifications and standards obtained by each enterprise in a province to describe regional food safety. The measurement of label and certification are defined by Equations (3) and (4).
(3)$$Label=\frac{Number\;of\;products\;with\;green\;food\;labels}{Number\;of\;enterprises\;with\;green\;food\;labels}$$
(4)$$Certification=\frac{Number\;of\;green\;food\;certified\;products}{Number\;of\;green\;food\;certified\;enterprises}$$
The characteristics of all variables are shown in Table 1.

### 2.3. Econometric Model

#### 2.3.1. Fixed-Effect Model

There are many variables that do not change with time. For example, social customs and culture affect the selection, production, storage, and consumption of food. Only following the traditional process to produce food and a lack of scientific production technology often lead to food exposure to harmful bacteria, affecting the incidence of FDEs in residents [49]. However, social customs and culture remain unchanged for a long time. However, the fixed-effect model can eliminate the influence of social customs on FDEs. So, the two-way fixed-effect model [50] was used to estimate the time-invariable factors affecting FDEs (Please notice that the Chinese government has regulatory authority over food at home and abroad. Therefore, no matter the food comes from home or abroad, the government can affect the food supply chain and then affect FDEs. Therefore, it is not necessary to distinguish whether FDEs comes from domestic markets or foreign market). The basic model is as follows:(5)$$y_{it}=\alpha +\beta x_{it}+u_{i}+\varepsilon_{it}$$

In Equation (5), $y_{it}$ denotes FDEs, α denotes the intercept, $x_{it}$ denotes the independent variable vector, β denotes the corresponding coefficient vector, $u_{i}$ denotes time-invariable variables, and $\varepsilon_{it}$ denotes error term.

#### 2.3.2. SYS-GMM

To avoid estimate bias by potential endogeneity, we added the lag terms of $y_{it}$ and $x_{it}$, which are the instrumental variables for endogenous variables GI in Equation (5). Then, Equation (5) transformed into Equation (6) below:(6)$$y_{it}=\alpha +\sum_{i=1}^{m}\sigma_{i}y_{i,t-i}+\sum_{j=0}^{n}\rho_{j}x_{j,t-i}+u_{i}+\varepsilon_{it}$$

In Equation (6), $\rho_{j}$ and $\sigma_{i}$ denote the corresponding coefficient of $y_{j,t-1}$ and $x_{i,t-1}$, and the meanings of other variables are the same as those in Equation (5).

#### 2.3.3. Spatial Econometric Model

For the spatiotemporal characteristics of FDEs in China, we used the spatial econometric model to capture the spatial effect of GI on FDEs. The spatial autocorrelation degree of FDEs was calculated using Moran’s index [51]. Because this study is focused on the relationship between FDEs in different geographical areas, the absolute longitude and latitude positions were chosen as the base for the provincial spatial weight matrix. The final Moran index is:(7)$$Moran’I=\frac{n\sum_{i=1}^{n}\sum_{j=1}^{n}w_{ij}\left(x_{i}-\bar{x}\right)\left(x_{j}-\bar{x}\right)}{\sum_{i=1}^{n}\sum_{j=1}^{n}w_{ij}\sum_{i=1}^{n}{\left(x_{i}-\bar{x}\right)}^{2}}$$

In Equation (7), $x_{i}$ and $x_{j}$ denote FDEs in area i and area j, $w_{i}{}_{j}$ denotes the derivatives of average geographical weight, and n denotes the number of areas.

If the Moran’s index is significant in China, the spatial econometric model should be employed. Moreover, there are three traditional spatial econometric models for research: SAR, SEM, and SDM (see Equations (8)–(10)).
(8)$$\mathrm{SAR}:\;y=\rho wy+x\beta +\varepsilon ,\varepsilon ∼N\left(0,\sigma_{\varepsilon }^{2}I_{n}\right)$$
(9)$$\mathrm{SEM}:\;y=x\beta +\mu ,\mu =\lambda w\mu +\varepsilon ,\varepsilon ∼N\left(0,\sigma^{2}I_{n}\right)$$
(10)$$\mathrm{SDM}:y=\rho wy+x\beta +wx\theta +\varepsilon ,\varepsilon ∼N\left(0,\sigma_{\varepsilon }^{2}I_{n}\right)$$

In Equations (8)–(10), y represents FDEs, x represents the independent variable vector, w represents weight (consistent with $w_{ij}$ of Equation (7)), ρ, μ, and θ denote the parameters to be estimated, and ε represents the error term. If ρ=0, SDM can be transformed into SEM; if θ=0, SDM can be transformed into SAR.

## 3. Results & Analysis

### 3.1. Effects of GI on FDEs

Table 2 shows the effects of GI on FDEs. From model (1)–(2), the results show that a 1% increase in GI causes a corresponding 1.3% decrease in FDEs. Generally speaking, FEDs may affect GI, and some crucial variables may be omitted in the model (1). So, the estimated coefficient of GI is biased. By employing the SYS-GMM method, we estimated the effect of GI on FDEs, and the coefficient was also negatively significant. These results indicate that our estimated GI coefficient is reliable.

An inverse U-shape relationship between per capita GDP and FDEs was captured by Zhang et al. [26], but the results show that the coefficient of the quadratic term is not significant in our model. The impact of urbanization on FDEs is significantly positive (*p* < 0.001). Ceteris paribus, a 1% increase in urbanization corresponds to a 0.43% increase in FDEs. The impact of average years of education on FDEs is significantly negative (*p* < 0.001). If people have been educated for one more year, FDEs decrease by 2.43% correspondingly. The impacts of CPI on FDEs are not significant, and price is not shown to be the key factor affecting FDEs in Chinese residents. The impacts of the Theil index on FDEs are also not significant, which highlights inequality is not the vital variable affecting China’s FDEs.

The effects of temperature, sunshine, and rainfall on FDEs were relatively small. Even if they are removed from the model, the R^2^ only ranges from 0.562 to 0.558 (see Model 1 in Table 3). When we only estimated the natural environment’s influence on FDEs in Model 3, their corresponding coefficient was still insignificant. The result show that the effect of socioeconomic factors on FDEs is more remarkable than natural environmental factors in China, which is not hard to understand. For example, the natural environment has an important impact on food storage in low-income areas. However, science and technology have changed the style of food storage, and food can be presented to consumers in a fresh state. So, in contrast, the impact of the natural environment becomes much weaker. Our findings are consistent with [43].

### 3.2. Effects of GI on Food Safety Standards

Table 3 illustrates the impact of GI on safe food production, which is the vital mechanism through which GI affects FDEs. Model (1)–(4) in Table 3 illustrate that GI has significantly increased the green food certification and labeling of enterprises. The results also show that GI promotes enterprises to carry out green food certification, improve regional food safety, and reduce FDEs.

### 3.3. Lagged Effects of GI on FDEs

It is imperative to maintain the policy effect. In Table 4, the lagged effect of GI on FDEs is captured. The results show that the impact of GI on FDEs will last for two years. When GI lags in one year, the impact of GI on FDEs will be at its maximum. However, after that, the effect will dissipate quickly. The results show that GI on FDEs should focus on long-term policies and avoid the loss of medical expenditure.

### 3.4. Effects of GI on FDEs among Different Areas

In Table 5, the effects of GI on FDEs considering area difference are captured. Overall, the impact of GI on FDEs in western, central, and inland areas is more apparent than that in the eastern and coastal regions. The eastern and coastal areas have a more perfect food industry system and a stricter regulatory system. In these areas, the incidence of FDEs has been relatively low. Although FDEs can also be reduced when GI is further strengthened, the effect of GI on FDEs is not significant. So, compared with the western and central regions, the impact of GI is relatively limited for eastern and coastal areas.

### 3.5. Spatial Autocorrelation Test

Moran’s index is used to test the spatial autocorrelation of FDEs from 2011 to 2019. Table 6 shows that the spatial autocorrelation between different areas is significantly negative. The results imply two issues. One is that FDEs in high-gathering locations will expand into low-gathering regions, which means FDEs are constantly extending outward when they break out. The other is that FDEs are more likely to occur in the low-gathering areas. Both situations indicate that there is a robust spatial autocorrelation in China’s FDEs. From 2011 to 2015, the spatial correlation of FDEs gradually increased. After 2015, the spatial correlation decreased slightly, but the effect was not obvious. The results show that the 2015 Food Safety Law of China might play an alleviating role in FDEs’ aggregation.

### 3.6. Spatial Spillover Effects of GI on FDEs

Because FDEs present spatial correlation in most years, it is necessary to consider the spatial econometric model to estimate the impact of GI on FDEs. Our tests showed that SDM is the better model (see Table 7 and Table 8).

Bear in mind that other studies did not fully consider the temporal and spatial characteristics of FDEs, while in this study, the spatial econometric model was used to further study the impact of GI on FDEs. In Table 8, SDM is used to improve the estimated effects of GI on FDEs. The results show that GI negatively affects FDEs in the main, direct, and total effect models. However, the indirect effect model shows that GI has a positive impact on FDEs. Thus, the results are beneficial for explaining the effects of GI on FDEs. In addition, the results show that GI suppresses local FDEs but increases FDEs in other regions. However, the spillover effect of GI on FDEs is smaller than the main effect. So, on the whole, GI can curb FDEs efficiently.

There are other variables with interesting characteristics. First, when the spatial econometric model as used, the relationship between per capita GDP and FDEs presented an inverse U-shape attribute (see column (1), column (2) and column (4)). The results show the necessity of using the spatial econometric model. However, per capita GDP does not have an obvious spatial spillover effect on FDEs (see column (3)). Secondly, according to column (1), column (2) and column (4), the results show urbanization has a positive effect on the occurrence of FDEs. However, column (3) shows urbanization has a negative spillover effect on the occurrence of FDEs in other areas. Finally, people with more years of education can help reduce the incidence of local FDEs (see column (1), column (2) and column (4)). However, people with more years of education can also increase FDEs in other areas (see column (3)).

## 4. Discussions

Firstly, the results of this study show that GI has an uncertain effect on FDEs. After considering the spatial spillover effect, the impacts of GI on FDEs are divided into the main effect and spillover effect. The main effect is significantly negative, while the spillover effect is significantly positive. It shows that GI reduces FDEs in local areas but increases FDEs in other areas. The results are reasonable. During the process of food production, circulation, and sales needs for linkages between regions and industries. A region must continue to strengthen its own food supervision capacity [35], but this may also cause pressure on the production of food enterprises, squeeze low-quality food enterprises, and reduce FDEs in the region. However, this promotes the transfer of low-quality food enterprises to other regions, further improving food safety risks and increasing FDEs in other regions accordingly.

Secondly, this study shows that there is an inverted U-shape relationship between economic development and FDEs. However, economic growth fails to produce an obvious spillover effect. The results show that the ability of economic cooperation among cities is too weak to restrict FDEs.

Thirdly, if the people with higher education increase, FDEs in this area is less but FDEs in other area is more. The result show that people with higher education levels are often concentrated in a certain region, which makes food knowledge in this region higher and further reduces the FDEs in this region. However, this also leads to the lack of food knowledge in other regions and further aggravates the FDEs in other regions [40]. Correspondingly, the FDEs in other regions are decreasing. 

Finally, the study shows that although urbanization increases the chance of local FDEs, it significantly inhibits FDEs in other regions. This implies that the agglomeration of population to a region aggravates the FDEs of the region. However, the agglomeration of a population to a region also indicates that population agglomeration in other regions is decreasing [39]. So, the effects of urbanization on FDEs in other areas is negative.

## 5. Conclusions

Using China’s provincial panel data from 2011 to 2019, this paper investigated the influence of GI on FDEs using OLS and the spatial econometric model. The results show that although GI has a positive effect on FDEs, the impact of GI on FDEs has an obvious negative externality. Education, urbanization, and other factors substantially impact FDEs, but we also need to pay attention to their spillover effect. The inhibitory effect of GI on FDEs in the central and western regions is greater than that in the eastern and coastal areas. This study can provide the following suggestions for Chinese policymakers.

First, although GI can inhibit local FDEs, GI significantly aggravates FDEs in other places. The results show that relying solely on the government to rectify FDEs can cause contradictions and conflicts between different regions. It is vital to coordinate the interests of different areas and various subjects. Therefore, it is imperative to promote regional food safety co-governance.

Second, the spillover effect of economic growth on FDEs is not obvious. So, it would be wise for Chinese government to continue to optimize the rate of economic development and promote high-quality economic development.

Third, people with more years of education increase the occurrence of FDEs in other areas. Policymakers should pay attention to the beneficial effect of the inflow of high-quality talents on the reduction in local FDEs and the deterioration effect of the outflow of high-quality talents on other FDEs. Policymakers could promote the education equity by strengthening food safety education in less-developed areas.

Final, excessive population size in urban areas increases the occurrence of FDEs in local areas. To curb FDEs, policymakers could appropriately and reasonably control the development of mega cities and big cities and accelerate the improvement of infrastructure in small- and medium-sized cities to avoid excessive population agglomeration in big cities, which affects FDEs and public health.

Our research has the additional contributions of explaining the relationship between GI and FDEs. However, there are still some limitations. First, for the GI index measurement, there is a need to find more accurate indicators. Second, further exploration of micro-mechanisms is needed to further explain the influence of the mechanism of GI and FDEs.

## Figures and Tables

**Figure 1 ijerph-18-13311-f001:**
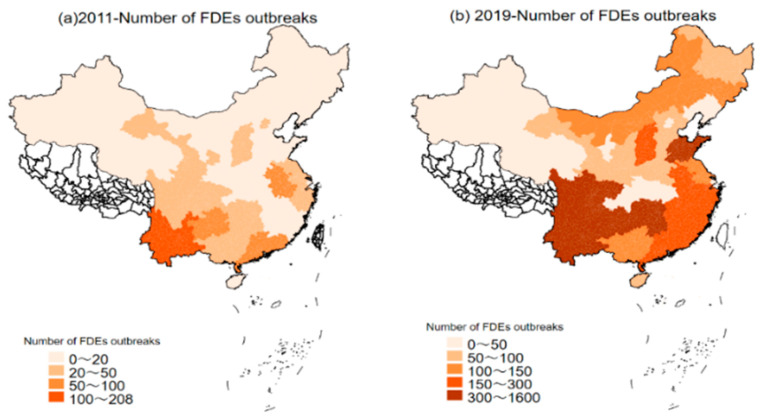
Spatial distribution of foodborne disease events from 2011 to 2019. Data source: China health statistics yearbook.

**Table 1 ijerph-18-13311-t001:** Description and definition of variables.

Variable	Definition	Obs	Mean	Std.Dev.
FDEs	(FDEs/Total population) ×100% ^a^	270	2.456	3.317
GI	Calculated by Equation (1)	270	2.524	1.269
Lpergdp	Ln (1+ GDP per capita (unit: yuan)	270	10.81	0.433
Urban	Proportion of registered residence in cities (unit: percentage)	270	57.64	12.18
Theil	Calculated by Equation (2)	270	0.0903	0.0809
Educ	Average education years of labor (unit: years)	270	9.533	0.815
CPI	Consumer price index (unit:%)	270	102.5	1.231
Sunshine	Ln (1 + average sunshine duration) (unit: hour)	270	7.591	0.217
Rainfalls	Ln (1 + average rainfall) (unit: 0.1 square millimeter)	270	9.072	0.501
Temperature	Ln (1 + average temperature) (unit: centigrade)	270	2.566	0.386
Label	Calculated by Equation (3)	210 ^b^	2.276	0.684
Certification	Calculated by Equation (4)	210	2.417	0.700

Notes: ^a^ FDEs represent the infection rate of foodborne diseases among ten thousand people. ^b^ At the time of the study, we were only able to collect the data of 21 provinces from 2011 to 2017 in China.

**Table 2 ijerph-18-13311-t002:** Effects of GI on FDEs.

	(1)	(2)	(3)	(4)
	FE_1	FE_2	FE_3	SYS-GMM
GI	−1.354 ***	−1.301 ***		−0.761 *
	(0.309)	(0.305)		(0.424)
Lpergdp	42.231	41.527		151.424
	(28.617)	(28.479)		(170.946)
lpergdp_2	−1.869	−1.844		−6.432
	(1.286)	(1.278)		(7.683)
Urban	0.434 ***	0.437 ***		−0.218
	(0.139)	(0.137)		(0.178)
Theil	1.830	2.414		−9.327
	(9.423)	(9.317)		(27.720)
Educ	−2.438 ***	−2.494 ***		−1.644 ***
	(0.662)	(0.652)		(0.374)
CPI	0.362	0.315		0.483
	(0.313)	(0.309)		(0.316)
Sunshine	2.610		2.193	−5.591
	(2.920)		(3.209)	(4.641)
Rainfalls	−0.555		−0.010	−5.327 **
	(1.335)		(1.447)	(2.556)
Temperature	−0.091		−6.801	2.726
	(4.297)		(4.197)	(4.265)
Year effect	YES	YES	YES	YES
Province effect	YES	YES	YES	YES
AR(2)				0.452
Hense				0.641
Constant	−286.573 *	−262.085 *	1.265	−816.306
	(160.422)	(157.420)	(33.295)	(959.582)
Observations	270	270	270	270
R-squared	0.562	0.558	0.432	-
Provinces	30	30	30	30

Notes: (1) The coefficient is the robust standard error from clustering to province; (2) *** denotes *p* < 0.01, ** denotes *p* < 0.05, * denotes *p* < 0.1; (3) All models controlled for time effect, province individual effect, and their interaction; (4) fixed-effect model in column (1), column (2), and column (3). Additionally, the corresponding superscript is FE_1, FE_2, and FE_3 separately. However, in the column (4), the SYS-GMM method is employed, and the corresponding superscript is SYS-GMM.

**Table 3 ijerph-18-13311-t003:** Effects of GI on green food label and certification.

	(1)	(2)	(3)	(4)
	Green Food Label		Green Food Certification	
GI	0.0414 ***	0.0195	0.0530 ***	0.0486 ***
	(0.0115)	(0.0119)	(0.0141)	(0.0151)
Control Variables	NO	YES	NO	YES
Year Effect	YES	YES	YES	YES
Provinces Effect	YES	YES	YES	YES
Observations	210	210	210	210
R-squared	0.102	0.241	0.104	0.190
Provinces	30	30	30	30

Notes: (1) The coefficient is the robust standard error from clustering to province; (2) *** denotes *p* < 0.01; (3) All models controlled for time effect, province individual effect, and their interaction.

**Table 4 ijerph-18-13311-t004:** Lagged effects of GI on FDEs.

	(1)	(2)	(3)	(4)	(5)
	NOLAG	LAG1	LAG2	LAG3	LAG4
GI	−1.354 ***	−1.430 ***	−0.454	0.337	−0.542
	(0.309)	(0.380)	(0.638)	(0.932)	(1.710)
Control Variables	YES	YES	YES	YES	YES
Year Effect	YES	YES	YES	YES	YES
Provinces Effect	YES	YES	YES	YES	YES
Observations	270	240	91	66	76
R-squared	0.562	0.549	0.586	0.525	0.331
Provinces	30	30	13	11	19

Notes: (1) The coefficient is the robust standard error from clustering to province; (2) *** denotes *p* < 0.01; (3) All models controlled for time effect, province individual effect, and their interaction.

**Table 5 ijerph-18-13311-t005:** Effects of GI on FDEs among different areas.

	(1)	(2)	(3)	(4)	(5)
	Western	Central	Eastern	Coastal	lnland
GI	−5.917 ***	−1.455 ***	−0.520	−0.670	−1.365 ***
	(1.508)	(0.453)	(0.431)	(0.492)	(0.446)
	(10.267)	(18.419)	(5.076)	(12.434)	(5.408)
Control Variables	YES	YES	YES	YES	YES
Year Effect	YES	YES	YES	YES	YES
Provinces Effect	YES	YES	YES	YES	YES
Observations	99	54	117	99	171
R-squared	0.649	0.875	0.634	0.647	0.589
Provinces	11	6	13	11	19

Notes: (1) The coefficient is the robust standard error from clustering to province; (2) *** denotes *p* < 0.01; (3) All models controlled for time effect, province individual effect, and their interaction.

**Table 6 ijerph-18-13311-t006:** Spatial autocorrelation test of FDEs based on Moran’s index.

Year	Moran’ I	Expectation	Standard Error	Z Statics	*p*-Value
2011	−0.129	0.034	0.131	−0.723	0.47
2012	−0.128	−0.034	0.139	−0.67	0.503
2013	−0.042	−0.034	0.14	−0.053	0.957
2014	−0.415	−0.034	0.143	−2.653 ***	0.008
2015	−0.533	−0.034	0.142	−3.513 ***	0.000
2016	−0.361	−0.034	0.143	−2.287 **	0.022
2017	−0.309	−0.034	0.142	−1.938 *	0.053
2018	−0.372	−0.034	0.142	−2.373 **	0.018
2019	−0.220	−0.034	0.143	−1.303*	0.096

Notes: *** denotes *p* < 0.01, ** denotes *p* < 0.05, * denotes *p* < 0.1.

**Table 7 ijerph-18-13311-t007:** Optimal model test for spatial econometric model.

H_0_	Hypothesis	Results ^a^	Conclusion
ρ=0	SEM is better than SDM	ρ≠0	SDM is better
θ=0	SAR is better than SDM	θ≠0	SDM is better

Notes: ^a^ Table 8 shows that ρ = −0.216 (*p* < 0.1), θ = 2.491 (*p* < 0.001), which holds that SDM is a better option than SAR and SEM.

**Table 8 ijerph-18-13311-t008:** Spatial effect of GI on FDEs.

	(1)	(2)	(3)	(4)
	Main Effect	Direct Effect	Indirect Effect	Total Effect
GI	−1.329 ***	−1.311 ***	0.238 *	−1.073 ***
	(0.278)	(0.269)	(0.129)	(0.237)
Lpergdp	44.526 *	45.915 *	−8.247	37.669 *
	(25.736)	(26.687)	(6.709)	(22.329)
lpergdp_2	−1.965 *	−2.030 *	0.364	−1.666 *
	(1.156)	(1.200)	(0.300)	(1.004)
Urban	0.414 ***	0.413 ***	−0.075 *	0.338 ***
	(0.125)	(0.124)	(0.044)	(0.106)
Theil	0.541	0.677	−0.055	0.622
	(8.494)	(8.607)	(1.731)	(7.077)
Educ	−2.455 ***	−2.448 ***	0.440 *	−2.009 ***
	(0.594)	(0.591)	(0.240)	(0.526)
CPI	0.351	0.353	−0.063	0.289
	(0.282)	(0.292)	(0.064)	(0.244)
Lrain	−0.595	−0.643	0.116	−0.528
	(1.199)	(1.119)	(0.230)	(0.931)
Ltem	−0.170	0.050	−0.011	0.038
	(3.860)	(3.778)	(0.743)	(3.118)
Lsun	2.443	2.439	−0.446	1.993
	(2.624)	(2.685)	(0.598)	(2.212)
ρ	−0.216 *			
	(0.122)			
θ	2.491 ***			
	(0.215)			
Observations	270			
R-squared	0.417			
Provinces	30			

Notes: *** denotes *p* < 0.01, * denotes *p* < 0.1.

## Data Availability

Not applicable. No new data were created or analyzed in this study.

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
