# Peer review of "Effects of Governmental Intervention on Foodborne Disease Events: Evidence from Chinaâ€"

_ijerph, 2021, doi:10.3390/ijerph182413311_

Round 1
Reviewer 1 Report
The manuscript is well structured and the approach is suitable for the aim. The study is of good relevance since food-borne diseases are an important cause of morbitidy and mortality, and a significant impediment to socio-economic development worldwide. Despite decades of government and industry interventions, food-borne disease remains unexpectedly high in both developed and developing countries.
However there are just few issues to be addressed:
- The section 2. Current situation and literature review - the literature is quite "old" and insufficient. Please add literature from a last couple of years. For example you can use data on cases reported through the National Foodborne Disease Outbreaks Surveillance system from 2020.
- The place of infection and impact of culture and eating behavior on food-borne diseases should be more discussed.
- Influenced by the COVID-19 pandemic, for the first time in the past five years, outbreaks that occurred in private homes surpassed those in catering service units in 2020. Therefore, the impact of COVID-19 pandemic on eating behavior should be discussed.
Author Response
Dear Reviewer,
Thanks for your careful suggestions! All our replies to the reviewers are attached below.
#Review1:
Q1: The manuscript is well structured and the approach is suitable for the aim. The study is of good relevance since food-borne diseases are an important cause of morbitidy and mortality, and a significant impediment to socio-economic development worldwide. Despite decades of government and industry interventions, food-borne disease remains unexpectedly high in both developed and developing countries.
#Reply: Thanks for your positive comments.
However, there are just few issues to be addressed:
Q2: The section 2. Current situation and literature review-the literature is quite "old" and insufficient. Please add literature from a last couple of years. For example, you can use data on cases reported through the National Foodborne Disease Outbreaks Surveillance system from 2020.
#Reply: Thank you for your valuable suggestion. To make our article more compact and more in line with the guidelines of the journal, we integrated the "2. Current situation and literature review" of the original manuscript into the "Introduction" part. (1) As you said, some of the documents I cited are “old”, so we cite some of the latest relevant studies. I have listed all these documents in “BLUE” in the “References” section.
(2) You are concerned that the data of “National Foodborne Disease Outbreaks Surveillance system” has not been updated to 2020. Your view is very reasonable. It is very sorry for that. We try our best to apply for the latest data, but only the data up to 2016 are open to access. As you suggest, the data is not "new" and easy to confuse readers. Therefore, we tried to delete "figure 1" in the original version. Of course, this may seem arbitrary, but it does not affect the reader's reading experience. If you have better suggestions, you can also discuss with us immediately.
Q3: The place of infection and impact of culture and eating behavior on food-borne diseases should be more discussed.
#Reply: Thank you for your suggestion. We further discussed the impact of food customs and culture on FDEs. But for that they do not change in the short term, and their impacts can be eliminated by fixed effect model.
You can look at the “Fixed effect model” in page 6 of this paper. “For example, social customs and culture affect the selection, production, storage and consumption of food. Only following the traditional process to produce food, lack of scientific production technology, often lead to food exposure to harmful bacteria, affecting the incidence of FDEs in residents.”
I hope this will make sense.
Q4: Influenced by the COVID-19 pandemic, for the first time in the past five years, outbreaks that occurred in private homes surpassed those in catering service units in 2020. Therefore, the impact of COVID-19 pandemic on eating behavior should be discussed.
#Reply: Your statement is very reasonable, but unfortunately, the Chinese government has not released the FDEs data in 2020, so we can't update the data to 2020. However, we have stated the relationship between COVID-19 and FDEs in the background. I hope this will make sense.
You can look at the first paragraph of the revised paper: “The residents' food safety and health have been challenged during the COVID-19 epidemic. Preventing FDEs is an important task for all countries for ensuring national security.”
Yours,
Zhang zhuang

Reviewer 2 Report
I think this paper has a lot to offer in terms of insight around the relationship between interventions and disease outbreak. For the most part, it was well-written and clear. However, I have concerns with many of the terms used that are often unnuanced and insensitive by English language standards-- I 100% presume this is unintentional. It requires a more thorough editing for this kind of language.
Here is my specific feedback:
Abstract langauge: "Should pay great attention to"-- much too strong of language and remove "great"--this is a value-laden adjective.
Introduction:
pagh 1:
- what do you mean by "vertical significance"?
- arugably it is not possible to "fully" consider the mechanisms of any complex intervnetion-- temper the language here
prgh 4:
- "awkward" seems an inappropriate word to use to describe spatio-temporal analsyis results--do you mean "limited" or...?
- take out summaries of your findings from the introduction or change the tense to be present tense.
Overall the introduction and justification is well done!
Lit reivew:
Pargh 7
- please change the term "economically backward" to describe lower-income or more informal economic regions-- this term of "backward" while I realise it is used alot in the South Asian and Asia Pacific regions, is not politically correct by international standards, as far as I know.
prgh 8:
- please add some nuance to your phrasing around ultra-poor peri-urban or urban slum-dwellers often eating stale food etc-- there is a reason this demographic often eats this kind of food and it is not by choice--how it is written makes it sound like this is their decision. More sensitivity required.
prgh 13:
- education-- define what you mean by "well-educated"...please be more objective. Does research show that those with x number of years of education experience less FDEs, for example?
Pgh 14
-again, what is meant by "vertical participants"?
pgh 16:
- same issue with how the authors have presented the idea of "better education" as the prgh 13 point above.
Figure 1: what is on the y axis? Incidence? Number of outbreaks? Also, can you make the dates more clear on the x-axis? and label both the axes?
Figure 2: I like this figure but can you put the year of each image right in the figure rather than just in the title?
Good thorough job on the literature review/synthesis!
Good description of the variables used.
Results:
- Good clear summary of the results of the FE models
- page 10-- another use of "economically backward"
- section 4.4-- please rephrase the proposal that people in this region have higher self-control-- this, in English, is full of value judgements and is inappropriate for academic writing.
-again, please change "backward"
page 13: what do you mean by "education can restrain local FDEs"? Do you mean that more years of education at the population level can help reduce the incidence of local FDEs? This is unclear
CONCLUSION and DISCUSSION
5.2: -- the second point about negative externalities--that first sentence of that paragraph needs to be rewritten-- this dangling participle makes it hard to understand.
-- the last point about "vigorously developing the economy"-- first, remove the word "vigorously" to describe this. Your recommendation here does not consider the other outcomes related to "vigorous" economic delveopemnt that could possibly have way worse effects than a bit of food poisoning, yes? What if you reframed this in terms of optimising the rate of economic delveopment?
- what do you mean by "controlling urbanisation" or devleoping education? Be specific and be transparent about your perspective here--in many contexts the phrase "controlling" urbanisation might be perceived as dictatorial or authoritarian.
Author Response
Dear Reviewer,
Thanks for your careful suggestions! All our replies to the reviewers are attached below.
Yours,
Zhangzhuang
#Review2:I think this paper has a lot to offer in terms of insight around the relationship between interventions and disease outbreak. For the most part, it was well-written and clear. However, I have concerns with many of the terms used that are often unnuanced and insensitive by English language standards-- I 100% presume this is unintentional. It requires a more thorough editing for this kind of language.
#Reply: Thanks for your helpful suggestion and we have improved the language accordingly.
Q1: Abstract langauge: "Should pay great attention to"-- much too strong of language and remove "great"--this is a value-laden adjective.
#Reply: Thanks for carefully remark! We have improved our Abstract.
“Abstract: Foodborne disease events (FDEs) endanger residents’ health around the world, including China. Most of the countries have formulated food safety regulation policies, but the effects of governmental intervention (GI) on FDEs are still unclear. So, this paper purposes to explore the effects of GI on FDEs by using Chinese provincial panel data from 2011 to 2019. The results show that: (i) GI has a significant negative impact on FDEs. Ceteris paribus, FDEs de-creased by 1.3% when government expenditure on FDEs increased by 1 unit. (ii) GI improves food safety and reduces FDEs by promoting enterprises to provide safer food. (iii) and GI has a strong negative externality. Although GI alleviates FDEs in local areas, it aggravates FDEs in other areas. (iv) Compared with the eastern and coastal areas, the effects of GI on FDEs in the central, western, and inland areas are more significant. GI is conducive to ensuring Chinese health and equity. Policymakers should pay attention to two tasks in food safety regulation. Firstly, they should continue to strengthen GI in food safety issues, enhance food safety certification, and strive to ensure food safety. Secondly, they should reinforce the collaborative governance of regional food safety issues and reduce the negative externality of GI.”
Introduction:
Q2: pagh 1
- what do you mean by "vertical significance"?
- arugably it is not possible to "fully" consider the mechanisms of any complex intervnetion-- temper the language here.
#Reply: Thanks for careful remark! We have improved our paper in page 2- 3.
“Although numerous studies have explored the relationship between government intervention (GI) and foodborne disease events (FDEs) or food safety via theoretical analysis, case analysis, and linear regression, the relationship is still ambiguous. But knowing the mechanism and efficiency of GI on FDEs has significant theoretical and practical significance. So, in this article, we further consider the mechanism of GI on FDEs. Moreover, considering the spatial characteristics of FDEs, we try to capture the effect of GI on FDEs via a spatial econometric model (Because the temporal and spatial correlation of FDES in China is very obvious, see Figure 1).”
Q3: prgh 4
- "awkward" seems an inappropriate word to use to describe spatiotemporal analsyis results--do you mean "limited" or...?
- take out summaries of your findings from the introduction or change the tense to be present tense.
#Reply: Thank you for your kind reminder.
(1) As you have emphasized, it is unreasonable to use the word "awkward" to describe the limitations of existing research. We have improved our paper in page 2-page 3. "Although numerous studies have explored the relationship between government intervention (GI) and foodborne disease events (FDEs) or food safety via theoretical analysis, case analysis, and linear regression, the relationship is still ambiguous. "
(2) Thank you for your valuable discussion on the tense of our paper. We rechecked the tense of the paper in several parts.
Introduction-page 3:
“So, in this article, we further consider the mechanism of GI on FDEs. Moreover, considering the spatial characteristics of FDEs, we try to capture the effect of GI on FDEs via a spatial econometric model (Because the temporal and spatial correlation of FDES in China is very obvious, see Figure 1). Our research shows that the impact of GI on FDEs critically depends on its negative externality. Only when the negative externality is strong enough will GI have a significant negative impact on FDEs. The major contribution of this paper is that it provides a new perspective on the impact of GI on FDEs.”
Methodology-page 3:
“The China Health Statistics Yearbook offers us with details about the number of foodborne disease events (FDEs) and health supervision (center) personnel. The China Statistical Yearbook provides us with details about per capita GDP, consumer price index, urban and rural disposable income, urban and rural population, and the corresponding information on education (illiteracy, primary school, junior high school, senior high school, junior high school, university, and graduate students). The National Meteorological Science Data (dataset of daily surface climatological data over China (V3.0)) offers us with details about light, temperature, and rainfalls in each province. The China Green Food Development Center offers us with details about green food certification and labeling information in each province.”
Conclusion-page 11
Using Chinese provincial panel data from 2011 to 2019, this paper investigates the influence of GI on FDEs. The results show that GI has a significant inhibitory effect on FDEs, and GI can further reduce FDEs by strengthening food safety standards. After considering the spatial spillover effect, the impacts of GI on FDEs are divided into the main effect and spillover effect. The main effect is significantly negative, while the spillover effect is significantly positive. It shows that the impact of GI on FDEs has an obvious negative externality. Education, urbanization, and other factors substantially impact FDEs, but we also need to pay attention to their spillover effect. The inhibitory effect of GI on FDEs in the central and western regions are greater than that in the eastern and coastal areas.
Overall the introduction and justification is well done!
#Reply: Thank you for your positive comments.
Lit reivew:
Q3:Pargh 7
- please change the term "economically backward" to describe lower-income or more informal economic regions-- this term of "backward" while I realise it is used a lot in the South Asian and Asia Pacific regions, is not politically correct by international standards, as far as I know.
#Reply: Thanks for your helpful suggestion and we have improved the language accordingly.
In page 3: “Existing studies show that the natural environment and socioeconomic factors have a crucial impact on the occurrence of FDEs. In general, FDEs in developed areas are affected by socioeconomic factors more seriously, while in less-developed areas, FDEs are affected by the natural environment heavily. Michael et al stressed the influence of economic and social factors on food safety and risk. People believe that food safety in an area is mainly affected by five factors: education, social network, social capital, family income, and unemployment rate.”
Q4:prgh 8
- please add some nuance to your phrasing around ultra-poor peri-urban or urban slum-dwellers often eating stale food etc-- there is a reason this demographic often eats this kind of food and it is not by choice--how it is written makes it sound like this is their decision. More sensitivity required.
#Reply: Thanks for your helpful suggestion, we reorganized related expressions in revised paper accordingly.
In page 4:
“Environment is also a key variable for FDEs. Adane et al. found that the environment, particularly high humidity and high temperature, are conducive to fungal reproduction, resulting in severe FDEs in sub-Saharan areas.”
Q5:prgh 13:
- education-- define what you mean by "well-educated"...please be more objective. Does research show that those with x number of years of education experience less FDEs, for example?
#Reply: Thanks for your helpful suggestion, we replaced "well-educated" with "people with higher education" in revised paper accordingly.
Q6:Pgh 14
-again, what is meant by "vertical participants"?
#Reply: Thanks for your helpful suggestion, we replaced "vertical participants" with " critical participants " in revised paper accordingly.
Page-1: “Governments, media, and consumers are critical participants in controlling FDEs. The media can provide production information of food companies, promote consumers’ understanding of the products, and alleviate the information asymmetry between consumers and producers. Education, knowledge and economic status of consumers affect the occurrence of FDEs.”
Q7:pgh 16:
- same issue with how the authors have presented the idea of "better education" as the prgh 13 point above.
#Reply: Thanks for your helpful suggestion, we replaced "better education" with " people with higher education" in revised paper accordingly.
Page-4: “First, people with higher education have a richer knowledge about food safety and will pay more attention to it. Second, people with higher education can share their knowledge of food storage, food nutrition, and health with other residents. ”
Q8: Figure 1: what is on the y axis? Incidence? Number of outbreaks? Also, can you make the dates more clear on the x-axis? and label both the axes?
#Reply: Thanks for your suggestions! We try to adjust the picture accordingly. You can view it in the revised version. Because the data of this picture is “old”, it may confuse the readers. We deleted the picture when we restructured the article. However, this does not affect our research content. I hope it make sense.
Figure 2: I like this figure but can you put the year of each image right in the figure rather than just in the title?
#Reply: Thanks for your suggestions! We have changed the Figure in revised paper.
The picture has been revised in page-2.
Good thorough job on the literature review/synthesis!
Good description of the variables used.
Q9:Results
- Good clear summary of the results of the FE models
- page 10-- another use of "economically backward"
#Reply: Thanks for your suggestion! We have replace "economically backward" with "low-income areas".
You can see in page 7.
“It is not hard to understand why. For example, the natural environment has an important impact on food storage in low-income areas. However, science and technology have changed the style of food storage, and food can be presented to consumers in a fresh state. So, in contrast, the impact of the natural environment has become much weaker. Our findings are consistent with Lake et al (2009).”
Q10: - section 4.4-- please rephrase the proposal that people in this region have higher self-control-- this, in English, is full of value judgements and is inappropriate for academic writing. -again, please change "backward".
#Reply: Thanks for your valuable suggestions. We I revised the vague language in details. You can see in page 8 of the revised paper:
“The eastern and coastal areas have a more perfect food industry system and a stricter regulatory system. In these areas, the incidence of FDEs has been relatively low. Although FDEs can also be reduced when GI is further strengthened, the effect of GI on FDEs is not significant. So, compared with the western and central regions, the impact of GI is relatively limited for eastern and coastal areas.”
Q11: page 13: what do you mean by "education can restrain local FDEs"? Do you mean that more years of education at the population level can help reduce the incidence of local FDEs? This is unclear.
#Reply: Thanks for your suggestion! We revised the ambiguous expression and replaced B with a. In addition, we further explained the meanings of "education can restrict local FDEs" accordingly. You can see in page 10:
“Finally, people with more years of education can help reduce the incidence of local FDEs but that has a FDEs increase effect for other areas. This shows that people with higher education are often concentrated in a certain region, which makes the level of food knowledge in this region higher and further reduces the FDEs in this region. However, this also leads to the lack of food knowledge in other regions and further aggravates the FDEs in other regions.”
CONCLUSION and DISCUSSION
Q12: 5.2-- the second point about negative externalities--that first sentence of that paragraph needs to be rewritten-- this dangling participle makes it hard to understand.
#Reply: Thanks for your suggestion. We improve it as following in page 11:
“Secondly, the negative spillover effect of GI on FDEs should also be concern. Our research shows that although GI inhibit local FDEs, but it intensifies FDEs in other places for the negative spillover effect.”
Q13: the last point about "vigorously developing the economy"-- first, remove the word "vigorously" to describe this. Your recommendation here does not consider the other outcomes related to "vigorous" economic development that could possibly have way worse effects than a bit of food poisoning, yes? What if you reframed this in terms of optimizing the rate of economic development?
#Reply: Thanks for your suggestion. Your suggestion is very reasonable. Economic development needs to pay attention not only to quantity but also to quality. Therefore, I changed "vigorous" economic development" to "optimize rate of economic development and promote high-quality economic development. ", which is very in line with China's actual situation. You can see in page 11:
“Finally, they should continue to make relevant supporting policies. For example, there is an inverted U-shaped relationship between economic development and FDEs. So, it is wise to continue to optimize rate of economic development and promote high-quality economic development.”
Q14:- what do you mean by "controlling urbanisation" or developing education? Be specific and be transparent about your perspective here--in many contexts the phrase "controlling" urbanisation might be perceived as dictatorial or authoritarian.
#Reply: Thanks for your suggestion. Your suggestions are very valuable. Our language “controlling urbanization” may lead to misunderstanding by some readers. So, we make our policy implications more specific. “Policymakers can appropriately and reasonably control the development of mega cities and big cities, and accelerate the improvement of infrastructure in small and medium-sized cities, to avoid excessive agglomeration of big cities affecting FDEs and public health” You can see page-11.
“In this study, the impact of price and inequality on FDEs is not significant, showing that the Chinese government has made great efforts in food price control and access. The Chinese government needs to promote these policies steadily in the future. They should also focus on the negative effects of urbanization and education on other areas. So, policymakers can appropriately and reasonably control the development of mega cities and big cities, and accelerate the improvement of infrastructure in small and medium-sized cities, to avoid excessive agglomeration of big cities affecting FDEs and public health. Then, they can also promote the education equity of interregional food knowledge and promote the integration of interregional food safety. It is expected that through the above measures, the negative impact of GI on other regions will be effectively avoided.”

Reviewer 3 Report
The aim of the paper was to explore the effects of GI on FDEs using Chinese provincial panel data from 2011 to 2019. Results could be of interest to some audiences however major changes are require.
- Authors must format the article according to the journal's guidelines. For example, the paragraphs:
“This paper concluded that GI reduces FDEs by strengthening the certification of safe food. More significantly, we found that GI has obvious negative externalities. That is, GI reduces FDEs in a region but in- creases FDEs in other places.
Our research enriches the relevant research on the relationship between GI and FDEs. On the one hand, our study shows that the effects of GI on FDEs are uncertain and depend on the externality of the GI. The results give enough explanations for the ambiguous relationship between GI and FDEs. On the other hand, we demonstrate that policymakers should adopt numerous measures to eliminate the externality of GI, pointing out that regional cooperation is an optimal choice.”
Are conclusions and should not be in the introduction part.
- The second part of the article “2. Current Situation and Literature Review” should be in the introduction. it is recommended to be more conscientious because the introduction is too long.
- The methodology must be reorganized according to the journal format.
Introduction
- What types of activities are recognized as government interventions (GI)?
- The following paragraphs should not be included in the introduction
“So, in this article, we fully considered the mechanism of GI on FDEs. Moreover, considering the spatial characteristics of FDEs, we tried to capture the effect of GI on FDEs via a spatial econometric model. Our research shows that the impact of GI on FDEs critically depends on its negative externality. Only when the negative externality is strong enough will GI have a significant negative impact on FDEs. The major contribution of this paper is that it provides a new perspective on the impact of GI on FDEs.”
“This paper concluded that GI re- duces FDEs by strengthening the certification of safe food. More significantly, we found that GI has obvious negative externalities. That is, GI reduces FDEs in a region but in- creases FDEs in other places.
Our research enriches the relevant research on the relationship between GI and FDEs. On the one hand, our study shows that the effects of GI on FDEs are uncertain and depend on the externality of the GI. The results give enough explanations for the ambiguous relationship between GI and FDEs. On the other hand, we demonstrate that policymakers should adopt numerous measures to eliminate the externality of GI, pointing out that regional cooperation is an optimal choice.”
- When mention that there are many case studies and theoretical studies, cite some of those and main findings.
Methodology
- What were the dependent and independent variables and the control variables?
- If the data was collected from China national surveys, how the observations included (270) were selected? Are these observations statistically representative?
- Why only “green certifications” were selected as a safety standard? What does it mean for a food to have a green certification?
- How is ensure that the food- borne disease events considered in this study were originated by a national food and not by an imported food?
Results
- No discussion of the results is presented
References
- References must be numbered in order of appearance in the text
Author Response
Dear reviewer,
Thanks for your careful review! All our replies to the reviewers are attached below.
Yours,
Zhuang Zhang
#Review3:The aim of the paper was to explore the effects of GI on FDEs using Chinese provincial panel data from 2011 to 2019. Results could be of interest to some audiences however major changes are require.
Q1: Authors must format the article according to the journal's guidelines. For example, the paragraphs:
“This paper concluded that GI reduces FDEs by strengthening the certification of safe food. More significantly, we found that GI has obvious negative externalities. That is, GI reduces FDEs in a region but in- creases FDEs in other places.
Our research enriches the relevant research on the relationship between GI and FDEs. On the one hand, our study shows that the effects of GI on FDEs are uncertain and depend on the externality of the GI. The results give enough explanations for the ambiguous relationship between GI and FDEs. On the other hand, we demonstrate that policymakers should adopt numerous measures to eliminate the externality of GI, pointing out that regional cooperation is an optimal choice.”
Are conclusions and should not be in the introduction part.
The second part of the article “2. Current Situation and Literature Review” should be in the introduction. it is recommended to be more conscientious because the introduction is too long.
#Reply: Thanks for your valuable suggestion! Your suggestion is wise and reasonable. We have changed the paper according to your suggestions and the journal’s guideline. “2. Current Situation and Literature Review” have been concluded in the introduction. And conclusions in introduction have been deleted.
Q2: The methodology must be reorganized according to the journal format.
Introduction
What types of activities are recognized as government interventions (GI)?
#Reply: Thanks for your suggestion! We have added the meanings of government interventions (GI) in page 1 and page 2.
“For example, government can affect food safety in production, circulation and sales by means of enacting food safety laws, issuing administrative orders and implementing economic penalties (which could be called government intervention (GI)), so as to affect the outbreak of FDEs.”
Q3: The following paragraphs should not be included in the introduction
“So, in this article, we fully considered the mechanism of GI on FDEs. Moreover, considering the spatial characteristics of FDEs, we tried to capture the effect of GI on FDEs via a spatial econometric model. Our research shows that the impact of GI on FDEs critically depends on its negative externality. Only when the negative externality is strong enough will GI have a significant negative impact on FDEs. The major contribution of this paper is that it provides a new perspective on the impact of GI on FDEs.”
“This paper concluded that GI re- duces FDEs by strengthening the certification of safe food. More significantly, we found that GI has obvious negative externalities. That is, GI reduces FDEs in a region but increases FDEs in other places.
Our research enriches the relevant research on the relationship between GI and FDEs. On the one hand, our study shows that the effects of GI on FDEs are uncertain and depend on the externality of the GI. The results give enough explanations for the ambiguous relationship between GI and FDEs. On the other hand, we demonstrate that policymakers should adopt numerous measures to eliminate the externality of GI, pointing out that regional cooperation is an optimal choice.”
#Reply: Thanks for your suggestion! We have integrated these paragraphs into “Introduction” section and “Conclusions and Discussion” section.
Q4: When mention that there are many case studies and theoretical studies, cite some of those and main findings.
#Reply: Thanks for your suggestion! We added references to illustrate that in revised paper. You can see page-2, “Although numerous studies have explored the relationship between government intervention (GI) and foodborne disease events (FDEs) or food safety via theoretical analysis [10], case analysis [25], and linear regression [26], the relationship is still ambiguous”.
Q5: Methodology
What were the dependent and independent variables and the control variables?
#Reply: Thanks for your question. In our study, FDEs is dependent variable, GI is independent variable, Pergdp, Urban, Inequality, Education, Price, Sunshine, Rainfalls and Temperature are control variables. To make it clearer, in “Methodology- Variables” section, we have explained the reasons for the selection of this variable and its classification in detail.
Q5: If the data was collected from China national surveys, how the observations included (270) were selected? Are these observations statistically representative?
#Reply: Thanks for your questions! We have added notes in page 3. There are 27 provinces and 4 municipalities directly under the central government in Chinese mainland. Besides Tibet, we have chosen all from 2011 to 2019. Therefore, the data in this article can represent the actual development of FDEs at the provincial level in China.
Q6: Why only “green certifications” were selected as a safety standard? What does it mean for a food to have a green certification?
#Reply: Thanks for your questions! We have added notes in page 6. We recognize that it is not appropriate to take the acquisition of green food labels or certificates as food safety standards, and we have revised it. Similar to the research of Yu et al. (2014), we believe that the number of green food labels or certificates obtained by the company can be regarded as an indicator of food safety.
Q7: How is ensure that the food-borne disease events considered in this study were originated by a national food and not by an imported food?
#Reply: Thanks for your questions. Actually, Chinese government has regulatory authority over food at home and abroad. Therefore, no matter the food comes from home or abroad, the government can affect the food supply chain and then affect FDEs. Therefore, it is not necessary to distinguish whether FDEs comes from domestic market or foreign market. We have added notes in page 6.
Q8: Results-No discussion of the results is presented
#Reply: Thanks! We strengthened the discussion in the results section.
First, we explain why “Our research shows that GI inhibit local FDEs, but it intensifies FDEs in other places for the negative spillover effect.” The corresponding reason is that “The process of food production, circulation and sales needs for linkages between regions and industries. A region need continues to strengthen its own food supervision capacity, but it may also cause pressure on the production of food enterprises, and squeeze low-quality food enterprises and re-duce FDEs in the region. However, this will promote the transfer of low-quality food enterprises to other regions, further improve food safety risks and increase FDEs in other regions accordingly.” It can be seen in page 10.
Second, we explain why “although urbanization increases the chance of local FDEs, it significantly inhibits FDEs in other regions”. The reason is that “This shows that the agglomeration of population to a region will aggravate the FDEs of the region. However, the agglomeration of population to a region also means that the degree of population agglomeration in other regions is decreasing. so, the FDEs in other regions are decreasing”.
Final, we explain why “Education inhibit local FDEs, but it intensifies FDEs in other places”. The reason is “People with higher education are often concentrated in a certain region, which makes the level of food knowledge in this region higher and further reduces the FDEs in this region. However, this also leads to the lack of food knowledge in other regions and further aggravates the FDEs in other regions.”
Q9: References-References must be numbered in order of appearance in the text
#Reply: Thank you! We have numbered the references in order of appearance in the text.

Round 2
Reviewer 3 Report
Even though the authors made most of the suggested changes, the article needs some corrections:
- Delete the last paragraph of the introduction as it is not necessary.
- Introduction should end with the paper objective. This is only mentioned in the abstract.
- Speak throughout the article in the third person. For example, after 2.1. Data it says, "To analyze the spatiotemporal characteristics and vital socioeconomic factors of FDEs, we used panel data collected from the China Health Statistics Yearbook, China Statistical Yearbook, National Meteorological Science Data and China Green Food Development Center." and it should say " To analyze the spatiotemporal characteristics and vital socioeconomic factors of FDEs, panel data was collected from the China Health Statistics Yearbook, China Statistical Yearbook, National Meteorological Science Data and China Green Food Development Center.
- In tables 2, 3, 4, 5, 6, 8 specify what superscripts mean.
- In the results section there are “some discussions” these should be move to their corresponding section.
- Discussion and conclusions sections should be separated.
- In the conclusion paragraph remove those sentences that are results such as “The results show that GI has a significant inhibitory effect on FDEs, and GI can further reduce FDEs by strengthening food safety standards” and, it is recommended to include those conclusions mentioned in the abstract.
- Policy implications should be part of the conclusions.
Author Response
Dear Reviewer,
Thanks for your careful suggestions! All our replies to the reviewers are attached below.
Even though the authors made most of the suggested changes, the article needs some corrections.
#Reply: Thanks for your positive comments.
Q1: Delete the last paragraph of the introduction as it is not necessary.
#Reply: Thank you for your valuable suggestion. We have deleted the last paragraph of original paper.
Q2: Introduction should end with the paper objective. This is only mentioned in the abstract.
#Reply: Thank you for your suggestion. We improve the objective in introduction section of this paper. I hope this will make sense.
You can see in page 3 of this paper.
“This study has a guiding significance for the Chinese government to change the supervision approach, strengthen regional food supervision cooperation, ensure residents' food safety and control FDEs.”
Q3: Speak throughout the article in the third person. For example, after 2.1. Data it says, "To analyze the spatiotemporal characteristics and vital socioeconomic factors of FDEs, we used panel data collected from the China Health Statistics Yearbook, China Statistical Yearbook, National Meteorological Science Data and China Green Food Development Center." and it should say " To analyze the spatiotemporal characteristics and vital socioeconomic factors of FDEs, panel data was collected from the China Health Statistics Yearbook, China Statistical Yearbook, National Meteorological Science Data and China Green Food Development Center.
#Reply: Your statement is very reasonable. We revised the languages in the third person throughout the article.
You can see paper-2:
“So, this article further considers the mechanism of GI on FDEs. Moreover, considering the spatial characteristics of FDEs, this study tries to capture the effect of GI on FDEs via a spatial econometric model (Because the temporal and spatial correlation of FDES in China is very obvious, see Figure 1).”
“To analyze the spatiotemporal characteristics and vital socioeconomic factors of FDEs, panel data are collected from the China Health Statistics Yearbook, China Statistical Yearbook, National Meteorological Science Data and China Green Food Development Center.”
You can see paper-4
“To reflect the consumer’s response in the market, CPI (consumer price index) is used to measure market price changes.”
“Then, China’s annual average sunshine, rainfall, and temperature data from 2011 to 2019 can be calculated.”
You can see paper-6
“If the Moran’s index is significant in China, the spatial econometric model should be employed”
You can see paper-7
“Table 2 shows the effects of GI on FDEs. From model (1)-(2), it shows that 1 % increase in GI causes a corresponding 1.3% decrease in FDEs.”
You can see paper-8
“In Table 4, the lagged effects of GI on FDEs are captured.”
“In Table 5, the effects of GI on FDEs by considering area difference are captured.”
You can see paper-9
“Moran’s index is used to test the spatial autocorrelation of FDEs from 2011 to 2019.”
You can see paper-10
“Bearing in mind that other studies did not fully consider the temporal and spatial characteristics of FDEs, while this study, spatial econometric model is used to further study the impact of GI on FDEs. In Table 8, SDM is used to improve the estimated effects of GI on FDEs.”
Q4: In tables 2, 3, 4, 5, 6, 8 specify what superscripts mean.
#Reply: Thank you for your suggestion. We further explain the meanings of superscripts in each table. “***denotes P <0.01, ** denotes P <0.05, * denotes P <0.1”, We hope it will make sense.
Q5: In the results section there are “some discussions” these should be move to their corresponding section.
#Reply: Thank you for your suggestion. We add Discussions section and move some important discussions into this section. We hope it will make sense.
Q6: Discussion and conclusions sections should be separated.
#Reply: Thank you for your suggestion.
Q7: In the conclusion paragraph remove those sentences that are results such as “The results show that GI has a significant inhibitory effect on FDEs, and GI can further reduce FDEs by strengthening food safety standards” and, it is recommended to include those conclusions mentioned in the abstract.
#Reply: Thank you for your suggestion. We further improved our abstract.
“Foodborne disease events (FDEs) endanger residents’ health around the world, including China. Most of the countries have formulated food safety regulation policies, but the effects of govern-mental intervention (GI) on FDEs are still unclear. So, this paper purposes to explore the effects of GI on FDEs by using Chinese provincial panel data from 2011 to 2019. The results show that: (i) GI has a significant negative impact on FDEs. Ceteris paribus, FDEs decreased by 1.3% when government expenditure on FDEs increased by 1 %. (ii) By strengthening food safety standards and guiding enterprises to offer more safety food, GI further reduce FDEs. (iii) However, GI has a strong negative externality. Although GI alleviates FDEs in local areas, it aggravates FDEs in other areas. (iv) Compared with the eastern and coastal areas, the effects of GI on FDEs in the central, western, and inland areas are more significant. GI is conducive to ensuring Chinese health and equity. Policymakers should pay attention to two tasks in food safety regulation. Firstly, they should continue to strengthen GI in food safety issues, enhance food safety certification, and strive to ensure food safety. Secondly, they should reinforce the co-governance of regional food safety issues and reduce the negative externality of GI.”
Q8: Policy implications should be part of the conclusions.
#Reply: Thank you for your suggestion. We put the Policy implications section into the conclusions section in the revised paper.
Yours,
Zhuang Zhang
